# *Caenorhabditis elegans* Deficient in DOT-1.1 Exhibit Increases in H3K9me2 at Enhancer and Certain RNAi-Regulated Regions

**DOI:** 10.3390/cells9081846

**Published:** 2020-08-06

**Authors:** Ruben Esse, Alla Grishok

**Affiliations:** Department of Biochemistry, BU Genome Science Institute, Boston University School of Medicine, Boston, MA 02118, USA; rmesse@bu.edu

**Keywords:** DOT-1.1/DOT1L, *Caenorhabditis elegans*, H3K79, H3K9, enhancer, chromatin, ChIP-seq

## Abstract

The methylation of histone H3 at lysine 79 is a feature of open chromatin. It is deposited by the conserved histone methyltransferase DOT1. Recently, DOT1 localization and H3K79 methylation (H3K79me) have been correlated with enhancers in *C. elegans* and mammalian cells. Since earlier research implicated H3K79me in preventing heterochromatin formation both in yeast and leukemic cells, we sought to inquire whether a H3K79me deficiency would lead to higher levels of heterochromatic histone modifications, specifically H3K9me2, at developmental enhancers in *C. elegans.* Therefore, we used H3K9me2 ChIP-seq to compare its abundance in control and *dot-1.1* loss-of-function mutant worms, as well as in *rde-4; dot-1.1* and *rde-1; dot-1.1* double mutants. The *rde-1* and *rde-4* genes are components of the RNAi pathway in *C. elegans*, and RNAi is known to initiate H3K9 methylation in many organisms, including *C. elegans*. We have previously shown that *dot-1.1(−)* lethality is rescued by *rde-1* and *rde-4* loss-of-function. Here we found that H3K9me2 was elevated in enhancer, but not promoter, regions bound by the DOT-1.1/ZFP-1 complex in *dot-1.1(−)* worms. We also found increased H3K9me2 at genes targeted by the ALG-3/4-dependent small RNAs and repeat regions. Our results suggest that ectopic H3K9me2 in *dot-1.1(−)* could, in some cases, be induced by small RNAs.

## 1. Introduction

Histone modifications play important roles in the regulation of gene expression by affecting chromatin compaction (e.g., histone tail acetylation) and by serving as recognition modules for chromatin remodelers and transcription regulators. The DOT1 (disruptor of telomeric silencing) histone methyltransferase (HMT), which targets lysine 79 (K79) in the globular region of histone H3, was first described in *Saccharomyces cerevisiae* [1,2], where it antagonizes yeast-specific heterochromatin factors. DOT1 homologs are conserved in metazoans [3], and H3K79 methylation follows transcription and correlates with the active transcription in higher organisms as well.

H3K79 methylation is much less understood compared to other histone modifications. However, a lack of DOT1 is lethal in animals, underscoring its essential biological function [4,5,6]. Although the majority of studies dedicated to mammalian DOT1-like (DOT1L) are focused on its oncogenic role in acute leukemias (reviewed in [7,8,9]), it is becoming clear that DOT1L has a profound impact on development (reviewed in [10]). It is essential for murine embryogenesis [6], proper cardiac function [11], and hematopoiesis [5,12].

Additionally, there is growing evidence that DOT1L plays an import role in neural development. Notably, neural tube defects in human fetuses have been associated with decreased levels of H3K79 methylation in the brain [13]. An investigation of DOT1L in murine models identified its significance for cerebral cortex development [14,15], specifically for proper neuronal specification and distribution among the six cortical layers [14]. The latter function of DOT1L is linked to its role in the proper expression of lineage-specific transcription factors and cell cycle regulators [14]. A conditional knock-out of *DOT1L* in a mouse cerebellum led to a reduced cerebellum size and ataxia symptoms [16]. This was associated with the deregulation of genes important for cell migration and axon guidance, as well as of genes encoding transcription regulators [16].

Although H3K79 methylation is a feature of all active genes, some developmentally-regulated genes, such as *HOXA* and *MEIS*, are especially dependent on DOT1L for their expression [17,18,19,20]. In the absence of DOT1L, *HOXA* genes accumulate the silent chromatin marks H3K27me3 [17,18] and H3K9me2 [17]. Moreover, the accumulation of H3K9me2 is dependent on reduced H3K9 acetylation under conditions of DOT1L suppression [17].

Our recent genomic analyses in *C. elegans* identified a significant enrichment of DOT-1.1 (*C. elegans* DOT1) and its partner ZFP-1 (AF10 in mammals) at enhancers associated with developmental genes, including, most notably, genes involved in neural development and function [4]. In addition, we demonstrated that such genes require *dot-1.1* and *zfp-1* function for their high expression [4]. Accordingly, we have observed neuronal phenotypes, such as defective neuronal migration, in *C. elegans zfp-1* reduction-of-function mutants [21]. Moreover, *zfp-1* has been implicated in regulating olfaction [22]. Therefore, the developmental phenotypes of animals lacking H3K79 methylation are most likely due to a reduced activation of enhancer-controlled lineage-specific genes. Consistent with the role of DOT1 in promoting lineage specification, *dot-1.1* reduction-in-function mutations have recently been isolated in a screen for inhibitors of developmental plasticity [23].

Molecularly, DOT-1.1 and ZFP-1(AF10) form a stable stoichiometric complex that is conserved from *C. elegans* to mammals [1,2,18,24,25,26] and safeguards an open chromatin configuration at the *HOXA* locus in mammals [17,18]. We, therefore, hypothesized that H3K79me methylation may be involved in protecting enhancers from heterochromatin.

In *C. elegans*, heterochromatin is associated with H3K9 methylation (reviewed in [27]). It is dispersed along chromosomes and most notably accumulates at the distal arms of autosomes [28]. In addition, small RNAs have been implicated in guiding H3K9me deposition in *C. elegans* (reviewed in [29,30]). Heritable H3K9 methylation at endogenous nuclear RNAi target genes is associated with the nuclear HRDE-1 Argonaute protein [31,32,33,34]. New studies revealed a complex relationship between siRNA-induced transcriptional silencing and siRNA-induced H3K9me3 deposition, which can be uncoupled [35]. Moreover, it was shown that RNAi-induced transcriptional silencing is a multi-step and multi-generational process that requires H3K9 methylation for the establishment of gene silencing, but not for silencing maintenance [36,37].

We have recently discovered that severe H3K79me deficiency in nematodes causes early lethality, which can be suppressed by the loss-of-function of the upstream-acting RNAi pathway genes *rde-4* and *rde-1* [4]. These results suggest that a H3K79me deficiency causes lethality through the enhanced activity of endogenous RNAi. Therefore, here we address two main questions: (i) whether H3K9me2 is elevated at enhancers when H3K79me is lost; (ii) if so, whether the H3K9me2 increase is dependent on *rde-1* and *rde-4* function.

We find that, indeed, there is an elevated cumulative deposition of H3K9me2 at enhancers, but not promoters, in *dot-1.1* null mutant worms. Although RDE-1 and RDE-4 are not responsible for this genome-wide cumulative increase, our results cannot exclude the possibility that some genes required for embryo viability are silenced in *dot-1.1(−)* animals in a nuclear RNAi- and RDE-1/4-dependent manner. Our initial analyses indicate that some repetitive elements and/or genes silenced by sperm-inherited small RNAs gain H3K9me2 in the absence of DOT-1.1 and may represent targets commonly regulated by DOT-1.1 (positively) and RNAi (negatively).

## 2. Materials and Methods

### 2.1. C. elegans Growth Conditions and Strains

Worms were maintained at 20 °C on standard nematode growth medium (NGM) agar plates spotted with *Escherichia coli* OP50. The following strains were obtained from the *Caenorhabditis* Genetics Center (CGC): wild type (WT) *C. elegans* N2 (Bristol), MT3002 (*ced-3(n1286)* IV), RB774 (*zfp-1(ok554)* III), VC40040, which contains *zfp-1(gk960739)* III, and VC40220, which contains *dot-1.1(gk520244)* I. The latter two strains were generated by the Million Mutation Project [38]. *zfp-1(gk960739)* and *dot-1.1(gk520244)* [4] were outcrossed from the parental strains at least four times. *zfp-1(ok554)* produces a truncated ZFP-1 protein deficient in DOT-1.1 interactions, and has been extensively characterized by our group [21,24,39,40]. *zfp-1(gk960739)* is a deletion that removes the first 109 amino acids from the long isoform of ZFP-1. It is characterized in [41]. *dot-1.1(gk520244)* contains a point mutation in the HMT domain of DOT-1.1 and was used in [4]. The partial loss-of-function alleles: *zfp-1(ok554)*, *zfp-1(gk960739), dot-1.1(gk520244)*, and *dot-1.1(gk105028)* show reduced H3K79me2 levels, as described in [4,24,41] and shown in Appendix A. WM49 (*rde-4(ne301)* III) [42] and WM27 (*rde-1(ne219)* V) [43] were obtained from the lab of Dr. Craig Mello. The *dot-1.1* null mutant strain COP1302 (*dot-1.1(knu339)* I; *ced-3 (n1286)* IV) was generated using a proprietary protocol by Knudra Transgenics and has been described earlier [4]. It lacks the DOT-1.1 methyltransferase and is severely depleted in terms of H3K79 methylation, as was shown by westerns in [4] and ChIP in Appendix A. The following double mutants were used: AGK783 (*dot-1.1(knu339)* I; *rde-1(ne219)* V) and AGK782 (*dot-1.1(knu339)* I; *rde-4(ne301)* III) [4]. To obtain synchronous larval stage 3 (L3) worms, hypochlorite-synchronized L1 stage animals were grown for 36 h post-hatching at 20 °C on standard NGM plates.

### 2.2. Evaluation of Gonad Migration Defects

To visualize gonads and record migration defects, live L4 stage worms were mounted in groups of 10 (i.e., 20 gonads scored on each slide) on 2% agarose pads in M9 buffer with 25 mM sodium azide and viewed at 400× magnification using Zeiss AxioImager Z1 (Feldbach, Switzerland). A number of abnormally positioned distal gonad arms was recorded (e.g., 5 out of 20). A total of ~100 to 250 gonads were scored for each strain.

### 2.3. Chromatin Immunoprecipitation

Synchronized L3 larvae were washed off the plates using an isotonic M9 solution. Fixation was performed for 30 min at 20 °C in crosslinking solution (2% paraformaldehyde in M9 buffer), and excess formaldehyde was quenched with 0.1 M Tris-HCl pH 7.5. The pellet was then washed twice with M9 buffer at 4 °C. The worm pellet was resuspended in 1 mL of ice-cold RIPA buffer (10 mM Tris-HCl pH 8.0, 0.1% SDS, 1% Triton X-100, 0.1% sodium deoxycholate, 1 mM EDTA, 0.15 M NaCl) supplemented with Halt™ Protease Inhibitor Cocktail (Thermo Fisher Scientific, Waltham, MA, USA). The suspension was then transferred to a 1 mL Covaris milliTUBE with AFA fiber, and shearing was performed using the following settings: peak incident power (PIP) = 240 W; duty factor = 20%; cycles per burst (cpb) = 200; time = 480 s. After shearing, the crude extract was spun for 10 min at 10,000× *g* and the supernatant was collected. The protein concentration was determined by the Bradford method. The extract corresponding to 1 mg protein was diluted to 500 μL with ice-cold RIPA buffer supplemented with Halt™ Protease Inhibitor Cocktail (Thermo Fisher Scientific). Antibodies specific for H3K9me2 (Abcam, Cambridge, UK, ab1220, 5 μg) were added and, after an incubation at 4 °C for 1 h, 50 μL of Dynabeads™ Protein G suspension (Thermo Fisher Scientific) was added and the mixture was incubated at 4 °C for 1 h on rotation. Beads were recovered with a Dynal magnetic particle concentrator (Invitrogen) and washed at 4 °C with: 0.5 mL buffer TSE-150 (0.1% SDS, 1% Triton X-100, 2 mM EDTA, 20 mM Tris-HCl pH 8.1, 150 mM NaCl), 2 × 0.5 mL buffer TSE-500 (0.1% SDS, 1% Triton X-100, 2 mM EDTA, 20 mM Tris-HCl pH 8.1, 500 mM NaCl), 0.5 mL buffer TSE-1M (0.1% SDS, 1% Triton X-100, 2 mM EDTA, 20 mM Tris-HCl pH 8.1, 1 M NaCl), 0.5 mL buffer LiCl (250 mM LiCl, 1% NP40, 1% deoxycholate, 1 mM EDTA, 10 mM Tris-HCl pH 8.1), and 2 × 0.5 mL buffer TE (10 mM Tris pH 8.0, 1 mM EDTA). Antibody-bound chromatin was recovered in 2 × 0.2 mL elution buffer (1% SDS, 0.1 M NaHCO_3_) by shaking at room temperature for 15 min. The elution buffer was added to input samples (5%) to a total volume of 0.4 mL. Input and ChIP chromatin samples were subsequently processed in parallel. After 20 μL of 5 M NaCl was added to each sample, crosslinks were reversed overnight at 65 °C. A total of 10 μL of 0.5 M EDTA, 20 μL of 1 M Tris-HCl pH 6.5 and 0.5 μL of 20 mg/mL Proteinase K (Thermo Fisher Scientific) was added and each sample was incubated for 1 h at 42 °C. DNA was recovered using the QIAquick PCR Purification Kit (Qiagen, Hilden, Germany) in accordance with the manufacturer’s instructions. The immunoprecipitated DNA was either quantified by quantitative real-time PCR to calculate the percentage of immunoprecipitation relative to the input DNA, as described previously [24], or processed for high-throughput sequencing. For the latter, immunoprecipitated DNA and input libraries were prepared using a TruSeq ChIP Library Preparation kit following the manufacturer’s instructions (Illumina, San Diego, CA, USA) and assessed on their concentration and quality on DNA chips with a Bioanalyzer (Agilent). A total of 75-bp single read sequences were generated on the NextSeq 500 sequencer according to manufacturer’s instructions (Illumina).

### 2.4. H3K9me2 ChIP-seq Data Pre-Processing and Read Mapping

Raw sequencing reads were first pre-processed by removing low-quality reads with Cutadapt software (v1.9.3) [44] and high-quality reads were aligned to the WS220/ce10 assembly of the *C. elegans* genome using QuasR (v1.16.0) [45] with default settings.

### 2.5. Coordinates of Enhancers, ATAC-seq Peaks, Promoters, ZFP-1 Peaks, ALG-3/4 Targets, Genomic Repeats, Chromosome Domains, and ERGO-1/RRF-3 Targets

The coordinates of genomic domains corresponding to enhancers were obtained from a previously published study [46]. These regions are characterized by the presence of enhancer signatures, such as the enrichment of the H3K4me1 and H3K27ac chromatin modification marks and a depletion of H3K4me3. ATAC-seq peaks were obtained from a previously published study [47]. It should be noted that ATAC-seq peaks often lie in regions of high enhancer activity and overlap with enhancer chromatin signatures [4]. As previously described [4], enhancer regions (defined either by chromatin signatures or by chromatin accessibility) at least 1500 bp distal to any annotated transcription or termination site were classified as distal enhancer regions, and those intersecting with the coordinates of annotated genes by at least 50 bp were considered intragenic enhancers. Promoter regions were defined arbitrarily as windows 1000 bp upstream and 500 bp downstream of each transcription start site. The ZFP-1 peaks were downloaded from GEO (accession GSE50301). The gene targets of the Argonautes ALG-3 and ALG-4 were obtained from [48], and those of ERGO-1 and RRF-3 were obtained from [49]. The coordinates of chromosome domains (tips, arms and centers) were obtained from [50]. Repeat regions were from [51]. The genomic coordinates of all the analyzed regions are available in Appendix A.

### 2.6. Comparison of Cumulative H3K9me2 and ZFP-1 ChIP-seq Signals at Genomic Regions and at Enhancer Regions

The comparison of the cumulative H3K9me2 and ZFP-1 ChIP-seq signals at genomic bins and at enhancer regions was completed by resorting to the deepTools suite (v3.3.2). ZFP-1 ChIP-seq coverage files were downloaded from GEO (accession GSE50301) and, for each pair of input and Immunoprecipitation (IP) samples, the ChIP-seq signal, expressed as the log2-transformed ratio between the value for the IP sample and that for the input sample, was calculated using the BigwigCompare tool. H3K9me2 signals (in WT worms) were calculated in a similar manner, except that the bamCompare tool was used. In order to evaluate how H3K9me2 and ZFP-1 are correlated genome-wide, the multiBigwigSummary utility was used to calculate the average scores for each ChIP-seq signal in 10,000 bp non-overlapping sliding windows spanning the genome, and then the plotCorrelation tool was used. The coverage of H3K9me2 and of ZFP-1 at enhancers and heatmap plotting were done using the computeMatrix and plotHeatmap tools. All regions were stretched or shrunken to the same length (1000 bp). A total of 500 bp windows were added upstream and downstream of each region.

### 2.7. H3K9me2 ChIP-seq Data Analysis

In order to ensure equal representation of each sample, all mapped reads were downsampled to the lowest sample using Sambamba (v0.6.5) [52]. For each sample pair (input and IP), regions of significant enrichment of H3K9me2 in the genome were determined using Mosaics-HMM (broad peaks) with the default parameters [53]. Venn diagrams showing overlaps of H3K9me2-enriched enhancers between the dot-1.1 null mutant and its background strain were generated using the Vennerable package in R. A consensus peak set was generated by merging all the peaks identified for each sample pair.

Read counting at genomic regions was performed with BEDtools MultiCovBed (v2.29.0). Only regions intersecting a H3K9me2 peak(s) were considered for subsequent analysis. After filtering out non-enriched regions, counts were normalized using the Trimmed Mean of M values (TMM) method, which takes the RNA composition bias into account, using the edgeR package (v1.16.0) [54]. ChIP-seq coverage at specific genomic regions was expressed as log2-transformed RPKM (reads per kilobase per million mapped) divided by the RPKM value of the corresponding input DNA sample (i.e., log2(ChIP/input)). The same numeric operation was performed for a region set comprised of 10,000 bp non-overlapping sliding windows spanning the genome. Log2-transformed fold changes between each mutant replicate and a background strain were then calculated for each region set and the values were represented in the form of cumulative distribution plots or boxplots. Two-sample Kolmogorov–Smirnov (for cumulative distribution plots) and Wilcoxon rank sum (for boxplots) tests were performed to compare the cumulative changes between each region set of interest and those in non-overlapping genomic bins spanning the genome. For ATAC-seq regions, heatmaps representing log2-transformed fold changes between each mutant replicate and the background strain were generated using the pheatmap package in R.

Genome-wide coverage files (bedGraph format) were generated using bamCompare (deepTools suite (v3.3.2)). The log2-transformed ratio between the ChIP sample and the corresponding input sample was calculated for each of the 10 bp windows spanning the genome. The raw sequencing data and coverage files have been submitted to the GEO database (submission Series GSE150407).

## 3. Results

To begin the investigation of chromatin changes associated with a depletion of H3K79me at enhancers in *C. elegans*, we performed a genome-wide analysis of the heterochromatic H3K9me2 modification in two sets of experiments using larval stage 3 (L3) worms. In the first set, we compared wild type worms with the partial loss-of-function mutant *zfp-1(ok554),* in which H3K79me is reduced [39,40,55]. In the second set, we compared the *ced-3(n1286)* background strain, the *dot-1.1(knu339); ced-3(n1286)* double mutant, in which the DOT-1.1-dependent H3K79me is lost [4], and two strains derived from *dot-1.1(knu339); ced-3(n1286)*; *dot-1.1(knu339); rde-4(ne301)* and *dot-1.1(knu339); rde-1(ne219)* [4]. We analyzed the H3K9me2 changes in *dot-1.1(knu339); rde-4(ne301)* and *dot-1.1(knu339); rde-1(ne219)* relative to the wild type strain as well as relative to *dot-1.1(knu339); ced-3(n1286)*.

### 3.1. H3K9me2 and ZFP-1/DOT-1.1 Occupy Different Genomic Locations

#### 3.1.1. Genome-Wide Anti-Correlation

First, we compared the H3K9me2 landscape of wild type L3 *C. elegans* (ChIP-seq data from this study) with the genome-wide localization of ZFP-1 at the same developmental stage (ZFP-1 ChIP-seq modENCODE data, GEO submission GSE50301). We note that initial ChIP-chip experiments using mixed embryos reported identical ZFP-1 and DOT-1.1 peaks genome-wide [24,40]. This is consistent with ZFP-1 and DOT-1.1 being present in stoichiometric quantities in a complex that does not include additional proteins [24], as well as with ChIP-qPCR data detecting both proteins at the same genomic locations in L3 worm extracts [24]. We have computed the genome-wide coverage of H3K9me2 and of ZFP-1 in 10,000 bp non-overlapping sliding windows and found that H3K9me2 and ZFP-1 occupy distinct regions in the *C. elegans* genome (Figure 1a).

#### 3.1.2. H3K9me2 and ZFP-1 at Developmental Enhancers

Next, we used the coordinates of enhancer domains, either predicted based on a specific histone modification signature [46] or based on the open chromatin configuration, annotated using ATAC-seq [47], to evaluate the H3K9me2 levels at these potential regulatory regions. We found a depletion of H3K9me2 (Figure 1b), which is consistent with its preferential localization at repetitive elements [51,56,57,58]. At the same time, ZFP-1 is enriched at enhancers (Figure 1b), as we reported earlier [4].

### 3.2. DOT-1.1 Loss Does Not Perturb the Preferential Localization of H3K9me2 to Autosome Arms

In *Saccharomyces cerevisiae*, changes in H3K79 methylation are known to affect telomeric chromatin [1,2,26]. Although *C. elegans* chromosomes differ from those of yeast and mammals due to their holocentric organization, H3K9me2/3 is notably enriched at autosomal arms as compared to central regions [28,57]. Nucleosomes containing the centromeric H3 variant, CENP-A, are also enriched within autosome arms and are in proximity to nucleosomes bearing H3K9 methylation [59]. Our H3K9me2 ChIP-seq data are consistent with these earlier findings. First, we computed the H3K9me2 coverage at each of 10,000 bp non-overlapping sliding windows spanning the genome, normalized the values for sequencing depth, and expressed them as the log2-transformed ratio between the IP sample and the input sample (see Materials and Methods). Then, we calculated the fold change between each mutant strain and the corresponding background strain. Each genomic bin was assigned as belonging to either a chromosome tip, arm or center, according to the coordinates obtained from [50]. We observed a similar enrichment of H3K9me2 at autosome arms in all tested strains (Appendix A). The similarity of H3K9me2 ChIP-seq profiles at regions enriched for this modification is also evident from genomic browser screenshots of the whole chromosomes (Appendix A) and H3K9me2 enrichment clusters (Figure 2).

Next, we performed additional quantitative analyses of the H3K9me2 changes at autosome arms between the strains of interest and control strains. Two-sample Kolmogorov–Smirnov (for cumulative distribution plots) and Wilcoxon rank sum (for boxplots) tests were performed to compare the cumulative H3K9me2 changes at autosome arms (Appendix A). The cumulative H3K9me2 changes at non-overlapping genomic bins spanning the genome where used as controls (Appendix A). We have not detected a pronounced difference between the control (wild type, *ced-3(n1286))* and *zfp-1* or *dot-1.1* mutant conditions. This indicates that H3K79me does not play a dramatic role in preventing H3K9me from spreading to the centers of autosomes.

### 3.3. Regions Corresponding to ZFP-1 Peaks, But Not Promoters, Gain H3K9me2 upon DOT-1.1 Loss

Our previous studies determined that ZFP-1 and DOT-1.1 binding is prevalent at the promoter and enhancer regions in the genome [4]. Given the genome-wide anti-correlation between the H3K9me2 and ZFP-1/DOT-1.1 locations (Figure 1a), we analyzed the cumulative H3K9me2 changes of the control and mutant strains, specifically at the promoters and genomic coordinates corresponding to ZFP-1 ChIP-seq peaks (modENCODE data) using the approaches described in Section 3.2.

Remarkably, we found no elevation in the H3K9me2 levels of the mutants at promoters (Figure 3a), whereas H3K9me2 was significantly elevated at the ZFP-1 peaks in strains with a *dot-1.1* null mutant background, but not in the *zfp-1* reduction-of-function mutant (Figure 3b). This could reflect a much more dramatic depletion of H3K79me in *dot-1.1*(knu339) [4] compared to *zfp-1(ok554)* [24]. Alternatively (or in addition), it is possible that at the conditions when only the truncated ZFP-1 protein, which lacks the DOT-1.1 interacting region, is produced from the *zfp-1(ok554)* locus [24,39], there is a compensatory mechanism that brings DOT-1.1 to chromatin at some specific locations. This latter possibility can explain a significant cumulative decrease in H3K9me2 at locations of ZFP-1 peaks in *zfp-1(ok554* (Figure 3b)).

Since DOT-1.1 depletion does not affect H3K9me2 at genes regulated by the DOT-1.1/ZFP-1 complex at promoters [24], and since ZFP-1 peaks are enriched in the enhancers, in addition to promoters [4], these analyses strongly suggest that: (1) the mechanisms of DOT-1.1/ZFP-1 function at the promoters and enhancers may be distinct, and (2) the H3K9me2 changes observed at ZFP-1 peaks in the *dot-1.1* null background correspond to enhancers.

### 3.4. H3K9me2 Is Elevated at Enhancers in dot-1.1(−) Worms

Indeed, our analyses of cumulative H3K9me2 signals over enhancer elements, as defined in Section 3.1 and in Materials and Methods, detected a significant elevation in this silencing mark in strains with a *dot-1.1* null mutant background (Figure 4a and Appendix A).

These data are consistent with the results described in Section 3.3 and with our published work [4,24]. Specifically, we highlighted the distinction between the promoter- and enhancer-regulated ZFP-1/DOT-1.1 target genes [4]. Promoter-regulated genes tend to be widely and highly expressed, and ZFP-1/DOT-1.1 negatively modulates the transcription of these genes by affecting the RNA polymerase II dynamics [24]. However, genes that show high levels of ZFP-1/DOT-1.1 binding throughout their bodies, including intragenic enhancers, are positively regulated by ZFP-1/DOT-1.1 [4]. This group is enriched in tissue-specific developmental genes, especially those involved in neural functions [4]. Notably, the mechanism of the ZFP-1/DOT-1.1-based “positive” regulation of developmental genes involves a suppression of antisense transcription [4]. We now find that the suppression of H3K9me2 is likely to contribute to the mechanism of developmental gene regulation by ZFP-1/DOT-1.1 as well. Our published work [4], together with the data presented here, strongly suggest a connection between an increased antisense transcription and ectopic H3K9me2 deposition at developmental genes in the absence of DOT-1.1. We provide illustrations of developmental genes gaining H3K9me2 in Figure 5 and Appendix A.

These include the *unc-44* gene coding for the ortholog of human ankyrin 2 and ankyrin 3, notably involved in the regulation of neural development (Figure 5a) [60,61,62,63,64,65,66,67,68], *sdpn-1*, which codes for synaptic dynamin binding proteins (Appendix A), the *egl-30* gene coding for G-protein subunit alpha q [69], which had been implicated in the control of acetylcholine release in neuromuscular junctions [70], the regulation of serotonin signaling controlling egg-laying [71,72], as well as in chemosensation [73] and axon regeneration (Appendix A) [74], and others. We have previously shown that *egl-30* [24] and *sdpn-1* [4] are regulated by ZFP-1/DOT-1.1.

Since mammalian DOT1L controls homeobox (HOX) genes, we evaluated *C. elegans* HOX homologs [75] and noted that DOT-1.1 binds to intronic enhancers of the *nob-1* gene (Figure 5b), one of the three *C. elegans* homologs of the posterior HOX gene Abd-B in flies and Hox9-13 in vertebrates [75]. *nob-1* is located in close proximity to another Abd-B-like gene, *php-3*. Importantly, H3K9me2 peaks were more frequent at the *php-3/nob-1* locus in *dot-1.1(−)*, as seen in Figure 5b. This suggests the conserved regulation of HOX genes by DOT1 from invertebrates to mammals, even though the number of the nematode HOX genes is reduced and they do not show typical linear arrangements [75].

Earlier, we have suggested that an elevated antisense transcription upon ZFP-1/DOT-1.1 loss-of-function may lead to ectopic dsRNA formation and the activation of the nuclear RNAi pathway, leading to heterochromatin formation at all gene body and/or enhancer regions bound by ZFP-1/DOT-1.1 [4]. This model was supported by our genetic data showing a suppression of the *dot-1.1* null mutant lethality by mutations in the RNAi components responding to dsRNA (*rde-4* and *rde-1*) [4]. However, the comparison of the cumulative enhancer H3K9me2 levels between *dot-1.1(−); ced-3(−)* and the strains lacking *dot-1.1* and either *rde-1* or *rde-4* did not detect reproducible H3K9me2 reductions (Figure 4b). These results do not exclude the possibility that elevated H3K9me2 in the *dot-1.1(−)* background at specific gene(s) regulating embryo viability is suppressed by the RNAi mutants. The possibility that enhancer H3K27me3 is elevated in *dot-1.1(−)* and that this elevation is dependent on RDE-1/4 in either a global or gene-specific manner also remains.

Next, we sought to determine the enrichment of H3K9me2 at enhancers in *dot-1.1(−); ced-3(−)* compared to *ced-3(−)* using a peak calling algorithm (see Materials and Methods). We counted the number of enhancers, defined either by ATAC-seq peaks [47] (Figure 6a,b) or by chromatin signatures [46] (Appendix A), that overlapped with the H3K9me2 peaks of both replicates for each strain. More H3K9me2-enriched enhancers were found in *dot-1.1(−); ced-3(−)* alone compared to the number of H3K9me2-enriched enhancers in *ced-3(−)* alone (Figure 6a,b and Appendix A). This indicates that the lack of DOT-1.1 drives the ectopic deposition of H3K9me2 at enhancers. This phenomenon is also illustrated by the screenshot of a large genomic region containing *unc-44* (Figure 5a): very few H3K9me2 peaks are observed in WT or *ced-3(−)* strains, but there is a visible increase in strains containing *dot-1.1(−)* and some indication of an increase in *zfp-1(ok554)*.

We generated heatmaps as an additional approach to visualize the H3K9me2 changes at ATAC-seq peaks [47] in the studied mutants. This representation shows clustering between the duplicate replicates, as well as clustering of *rde-4(−); dot-1.1(−)* with the two replicates of *rde-1(−); dot-1.1(−)* (Figure 6c,d). The latter is fully consistent with the known molecular connection between RDE-4 and RDE-1 [42]. The heatmap representation also highlights a cluster of intragenic ATAC-seq regions with common H3K9me2 increases in *dot-1.1(−); ced-3(−), rde-1(−); dot-1.1(−),* and *rde-4(−); dot-1.1(−)*, as seen in Figure 6c (middle). Notably, there is also a cluster of genes with elevated H3K9me2 levels in *dot-1.1(−); ced-3(−)* that does not show a similar increase in *rde-1(−); dot-1.1(−)* and *rde-4(−); dot-1.1(−)*, as seen in Figure 6c (top). This is consistent with the possibility that *rde-1(−)* and *rde-4(−)* suppresses the elevated H3K9me2 levels seen in *dot-1.1(−)* at some loci. Although H3K9me2 changes in *dot-1.1(−); ced-3(−)* cluster with those in *zfp-1(ok554)* in the case of intragenic ATAC-peaks (Figure 6c), they appear most dissimilar in gene-distal ATAC-seq locations (Figure 6d). The opposite effects of *dot-1.1* null and *zfp-1(ok554)* mutations can be explained by a possible compensation of the lack of DOT-1.1 recruitment by ZFP-1 in *zfp-1(ok554)* by a DOT-1.1 recruitment via YEATS domain-containing protein, GFL-1 [76]. GFL-1 is a homolog of the known mammalian interactors of DOT1L, ENL and AF9 [25,77]. Although this possibility remains to be tested experimentally, we note similar phenotypes of *zfp-1(RNAi)* and *gfl-1(RNAi)* nematodes with respect to vulva development and response to dsRNA [76,78].

Overall, the data presented here strongly implicate DOT-1.1 in the regulation of developmental enhancers in nematodes and suggest the involvement of the RNAi pathway in enhancer regulation in some cases.

#### *dot-1.1* and *zfp-1* Mutant Worms Exhibit Gonad Migration Defects Characteristic of UNC-6/Netrin Signaling Mutants

Given the global role of DOT-1.1 in enhancer and promoter regulation, it is not surprising that DOT-1.1 complex mutants show pleiotropic phenotypes. Earlier, we have implicated ZFP-1/DOT-1.1 in the control of life span and stress resistance [40], as well as neuronal migration [21], due to the direct regulation of the promoter of the insulin signaling pathway kinase gene *pdk-1* [24,40]. The early lethality of *dot-1.1(knu339)* worms is ultimately dependent on the apoptotic caspase CED-3 [4], but it is not yet clear which DOT-1.1 target genes are responsible for this phenotype. A new investigation of the viable *dot-1.1(knu339); ced-3(n1286)* strain found multiple abnormalities in meiosis and oocyte development [41], with key regulator genes dependent on DOT-1.1 not yet identified.

A notable developmental phenotype associated with the available partial loss-of-function *zfp-1* and *dot-1.1* mutants (Figure 7a, Table 1) and independently identified in RNAi screens [79] is an abnormal gonad shape that indicates defects in ventral–dorsal gonad migration. This process is regulated by netrin signaling in *C. elegans* [80]. The secreted guidance cue netrin (UNC-6 in *C. elegans*) and its receptors UNC-5 and deleted in colorectal carcinoma (DCC (UNC-40 in *C. elegans*)) have a critical role in neural development and in the morphogenesis of a number of organs and tissues [81]. The profound defects in developmental cell migrations and axon guidance in *unc-6*, *unc-5* and *unc-40* mutants were first described in *C. elegans* [80]. UNC-6 is expressed in several types of ventral cells in *C. elegans* larvae and stimulates attractive or repulsive migrations along the ventral–dorsal axis of the animal [82] (Figure 7a, top). In loss-of-function netrin signaling mutants, the dorsal migration of the gonads often fails, resulting in a ventral position of the gonad and a dorsal displacement of the intestine [80] (Figure 7a, bottom). We observed this defect in *zfp-1(gk960739)* and *dot-1.1(gk520244)* mutants (Figure 7a, bottom, Table 1). The defects are much more penetrant in *dot-1.1(gk520244* (44% defective gonads, *n* = 166)) compared to *zfp-1(gk960739* (5%, *n* = 232)). In *dot-1.1(gk520244)*, a posterior ventralized gonad appearance is most common (71%, *n* = 83); similarly, posterior gonads are more affected than anterior gonads in *unc-6*, *unc-5* and *unc-40* mutants [80].

Notably, the *unc-6* gene contains the predicted enhancers in its introns (Figure 7b), and DOT-1.1 is bound throughout >5000 bp of *unc-6* in mixed embryo preparations (Figure 7b) [24]. This localization is consistent with a model in which DOT-1.1 and H3K79me occupy *unc-6* and other developmental genes and allow their tissue- and cell-specific activation by lineage-specific transcription factors through safeguarding the permissive open chromatin. Although we used larval stage 3 worms in our H3K9me2 ChIP-seq experiments, the absence of *dot-1.1* at this stage led to a noticeable increase in H3K9me2 peaks at *unc-6* intronic enhancers (Figure 7b). Our results are consistent with the possibility that a reduced *unc-6* expression results in an abnormal gonad migration in *dot-1.1(gk520244)* mutants and that ectopic H3K9me2 contributes to decreased *unc-6* expression.

### 3.5. H3K9me2 Is Elevated at Repetitive Elements in dot-1.1(−) Worms

Since H3K9me2 enrichment at chromosome arms is not affected by DOT-1.1 loss (see Section 3.2), we were surprised to detect an elevation in cumulative H3K9me2 signals at repeat regions in animals carrying *dot-1.1(−)*, most notably when *dot-1.1(−); rde-1(−)* was compared to WT (Appendix A). The elevation driven by *dot-1.1(−)* was apparently suppressed by *rde-4(−).* The best characterized targets of the heritable endogenous nuclear RNAi silencing are retrotransposons [31,32,35,36]. Therefore, we surveyed the specific examples of the Long Terminal Repeat (LTR) loci regulated by HRDE-1-dependent H3K9me described by Gu and colleagues [35,36] and found increased H3K9me2 levels at the Cer8 retrotransposon in *dot-1.1(−); ced-3(−)* and *rde-1(−); dot-1.1(−)* animals (Figure 8).

Although we cannot draw conclusions about the role of RDE-4 in the cases where it appears to act differently from RDE-1 in our experimental set-up, an increase in H3K9me2 at repeat regions detected when *dot-1.1(−); rde-1(−)* is compared to *dot-1.1(−); ced-3(−)*, as seen in Appendix A, suggests that the lack of RDE-1 function may remove its competition with some Argonaute proteins promoting LTR silencing. Overall, these results suggest that DOT-1.1 may counteract nuclear RNAi-dependent H3K9me depositions at repetitive elements in *C. elegans*.

### 3.6. H3K9me2 Is Elevated at ALG-3/4 Target Genes in dot-1.1(−) Worms

The endogenous RNAi system is extensive in *C. elegans* with ~20 Argonaute proteins present [83]. Only some endo-RNAi pathways start with the dsRNA cleavage by the Dicer complex [49,84]. Since RDE-1/4 functions with Dicer, we sought to survey the changes in H3K9me2 at the known target genes of Dicer-dependent small RNAs, 26G-RNAs [48,49,84]. There are two established 26G-RNA pathways driven by the Argonaute proteins binding to them: the EGRO-1 pathway [49] and the ALG-3/4 pathway [48]. The former acts in the oocytes and embryos and the latter during spermatogenesis [48,49,84]. The target genes regulated by either of the pathways were defined based on the depletion of 26G-RNAs, matching the genes in *ergo-1(−)* [49] or *alg-3/4(−)* [48] nematodes. We have not detected H3K9me2 changes at the EGRO-1 target genes in animals lacking DOT-1.1 (Appendix A). However, the ALG-3/4 target genes showed elevated levels of H3K9me2 when DOT-1.1 was missing (Figure 9).

The comparison between *dot-1.1(−); ced-3(−)* and *rde-1(−); dot-1.1(−)* did not show H3K9me2 elevation at the ALG-3/4 targets (Appendix A), in contrast to the repeats (Appendix A). Moreover, by surveying H3K9me2 browser tracks we found examples of elevated H3K9me2 in *dot-1.1(−); ced-3(−)* that was not detected in *rde-1(−); dot-1.1(−)* or *rde-4(−); dot-1.1(−),* as seen in Appendix A. Notably, the ALG-3/4 targets include many genes essential for embryonic and larval development (*let*, lethal), such as *let-70*, *let-721*, and *let-754* [48]. The *icd-1* (inhibitor of cell death) gene [85] also belongs to the ALG-3/4 target list [48]. Intriguingly, the *icd-1(−)* embryos are notable for the abundance of vacuoles in the intestinal cells [85], and *dot-1.1(−)*-arrested larvae show prominent vacuoles in the intestine as well (Figure 10).

Therefore, the analyses presented here suggest the possibility that ALG-3/4-dependent small RNAs, in addition to regulating spermatogenesis-specific genes [48], inhibit the expression of some widely-expressed essential genes in the developing sperm. At the same time, DOT-1.1 acting in the somatic cells may counteract silencing imposed by such heritable small RNAs on genes essential for embryo and larva development.

## 4. Discussion

Here we provide evidence that, in the absence of *C. elegans* DOT-1.1, the heterochromatic H3K9me2 modification increases at enhancer, but not promoter, elements. Because DOT-1.1 and its interacting partner ZFP-1 are themselves localized to both promoters and enhancers [4], our results are consistent with the distinct functions of the DOT-1.1 complex at these elements. Indeed, we have described the negative modulation of pol II transcription by promoter-localized ZFP-1/DOT-1.1 in developing larvae [24]. This mechanism is responsible for the lower expression of growth-promoting and metabolic genes in differentiated somatic cells and also contributes to gene regulation in response to environmental changes [24,40]. Now, not surprisingly, we find that H3K9me2 regulation is not a part of this mechanism. On the contrary, enhancer-localized ZFP-1/DOT-1.1 is responsible for the proper expression of developmental genes [4], and the mechanism in this case appears to be similar to that described for HOX genes in mammalian cells and involves regulation of H3K9 methylation.

Our findings suggest a direct role of DOT-1.1 in preventing H3K9me2 deposition at enhancers in *C. elegans*. We believe that H3K79 methylation deposited by DOT-1.1 is responsible for the observed effect, although we cannot exclude histone methyltransferase-independent functions of DOT-1.1. Additional experiments are required to prove this. Supporting evidence for the role of H3K79me in heterochromatin regulation exists in leukemic cells, where increased H3K9me2 levels at the *HOXA* locus, upon the chemical inhibition of DOT1L HMT functions, occurs due to Sirtuin 1 (SIRT1)-mediated H3K9 deacetylation, followed by a H3K9me2 deposition by histone methyltransferase SUV39H1 [17]. In addition, DOT1L inhibitor-induced losses of H3K79me at enhancers in leukemia cells has been correlated with reduced levels of H3K27 acetylation and reduced chromatin accessibility (probed by ATAC-seq) [86,87]. These lines of evidence suggest that H3K79 methylation may inhibit the chromatin localization of histone deacetylases and/or promote the function of histone acetyltransferases.

Our study demonstrates the antagonism between DOT1 and H3K9me2 at enhancers of developing animals for the first time, to the best of our knowledge, although specific enhancers enriched in H3K79me have been described in leukemia cells [86,87]. Interestingly, a recent quantitative study of gene expression plasticity (GEP), i.e., the capacity for large degrees of change with changing conditions, identified conserved histone modification signatures that affect plasticity in metazoans (including humans, mice, *Drosophila* and *C. elegans*) [88]. This study found that i) high levels of H3K79me at gene bodies restricted GEP and ii) homeobox genes, such as *HOXA*, had reduced plasticity [88]. These results are consistent with the role of DOT1 and H3K79me2 in ensuring proper animal development despite changing environmental conditions through the regulation of enhancers. Moreover, the identification of *dot-1.1* mutants in screens for inhibitors of cellular plasticity in *C. elegans* [23] further supports its role in developmental enhancer control.

An important question that we sought to address is the relationship between small RNA-induced gene silencing, heterochromatin formation and DOT-1.1 function. We have recently reported that genes harboring large domains of ZFP-1/DOT-1.1 binding at coding regions, including intragenic enhancers, decreased in expression in *zfp-1(ok554)* mutant worms [4]. This was accompanied by elevated levels of antisense transcription, suggesting that ZFP-1/DOT-1.1 regulates enhancer-containing genes by suppressing antisense transcription [4]. Although an elevation of ncRNA transcription and the deposition of silencing chromatin marks may appear to be mutually exclusive, the connection between the two phenomena is well documented in the case of RNAi-induced transcriptional silencing, where dsRNA initiates H3K9me (reviewed in [29,89]). Notably, we also found that *C. elegans* mutants deficient in the dsRNA-responsive pathway, *rde-4* and *rde-1*, suppressed the lethality of *dot-1.1(knu339)* [4]. Therefore, we considered the possibility that in the absence of DOT-1.1, ectopic dsRNA generation (due to sense/antisense hybridization) from intragenic enhancers may promote H3K9 methylation. The ChIP-seq results presented in this study indicate that the cumulative increases in H3K9me2 levels at enhancers observed in *dot-1.1(knu339)* are not diminished by *rde-1* or *rde-4* loss-of-function mutations. However, our results do not exclude the possibility of the antagonistic relationship between DOT-1.1 and endogenous RNAi at a subset of enhancer-regulated genes or other genomic elements. Indeed, we observed increased H3K9me2 levels at the repeat elements, including retrotransposones previously shown to be regulated by heritable nuclear RNAi [35,36]. Moreover, we found that genes targeted by heritable small RNAs of the ALG-3/4 pathway [48] show elevated H3K9me2 levels in *dot-1.1(−)* larvae. There are a number of genes essential for embryonic and larval development in this category, and we speculate that elevated levels and/or activity of ALG-3/4-dependent 26G small RNAs are responsible for the early lethality of *dot-1.1(−)* animals. If this is the case, it will be important to investigate the connection between ALG-3/4 and RDE-1 functions. RDE-4 has already been linked to pathways producing 26G-RNAs, a class of endogenous small RNAs produced by the Dicer complex [83].

Our results suggest that there must be another (and more prevalent) link between increased antisense transcription and H3K9me2 deposition at enhancers upon DOT-1.1 loss, in addition to ectopic RNAi activation. Studies of the mammalian *HOXA* locus identified an elevation in H3K27 methylation, a mark of a facultative chromatin, in addition to H3K9me2, when DOT1L was inhibited [18]. Importantly, antisense transcripts have been connected to H3K27me3 deposition and gene silencing at the cold-responsive locus FLC (flowering locus C) in plants [90,91]. Moreover, an interaction between the Polycomb group complex, which deposits H3K27me3, and RNA was shown *in vitro* [92], and the functional interconnection of RNA/H3K27me is an active area of investigation (reviewed in [93]). In *C. elegans,* we recently detected a dramatic increase in H3K27me3 at normally highly-expressed genes concomitantly with their reduced expression in conditions when antisense transcription was significantly elevated [94]. It is possible that increased antisense transcription leads to high H3K27me3 levels at enhancer elements in *dot-1.1(knu339)* as well, and that H3K27me3 and H3K9me2 both contribute to the reduced expression of developmental genes in animals lacking DOT-1.1.

## Figures and Tables

**Figure 1 cells-09-01846-f001:**
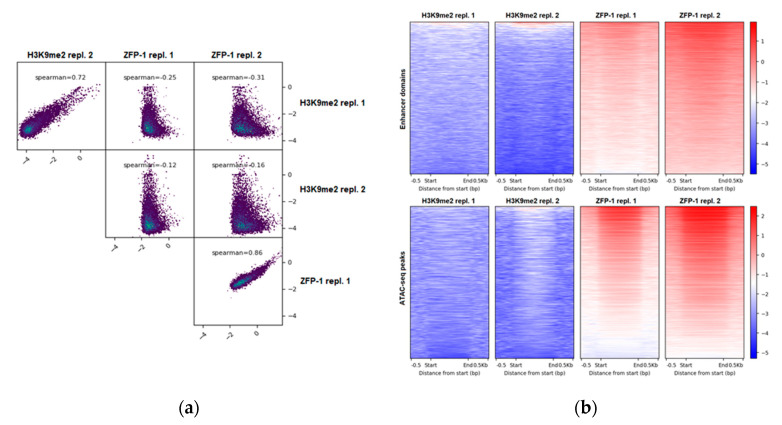
H3K9me2 and ZFP-1/DOT-1.1 occupy different genomic locations. (**a**) H3K9me2 (our data) and ZFP-1 (modENCODE data, GEO submission GSE50301) anti-correlate genome-wide. ChIP-seq signals (see Materials and Methods) were computed at 10,000 bp in non-overlapping sliding windows spanning the genome. Spearman correlations were calculated using the plotCorrelation tool (deepTools suite (v3.3.2)). (**b**) H3K9me2 is depleted, whereas ZFP-1 is enriched at enhancers. Coverages were calculated for either enhancer domains (top) [46] or ATAC-seq peaks (bottom) [47], and heatmaps were plotted using the plotHeatmap tool (deepTools suite (v3.3.2)).

**Figure 2 cells-09-01846-f002:**
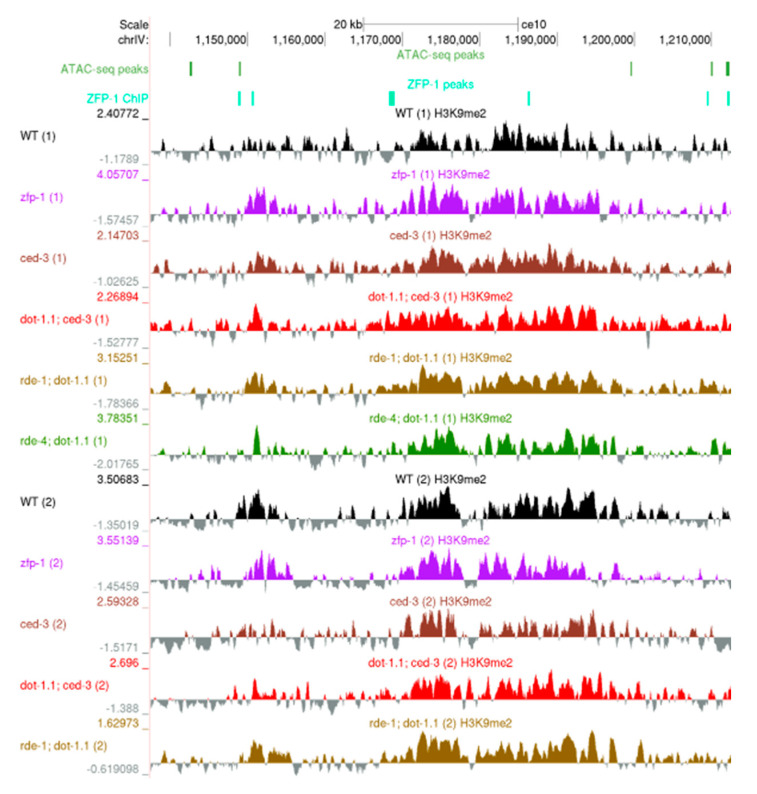
A University of California, Santa Cruz (UCSC) genome browser screenshot showing a representative ~70 kb region on the left arm of ChrIV enriched in H3K9me. ATAC-seq peaks: [47]. ZFP-1 peaks: modENCODE data, GEO submission GSE50301. H3K9me2 coverage tracks: our data; all samples are shown; replicates are marked by (1) and (2).

**Figure 3 cells-09-01846-f003:**
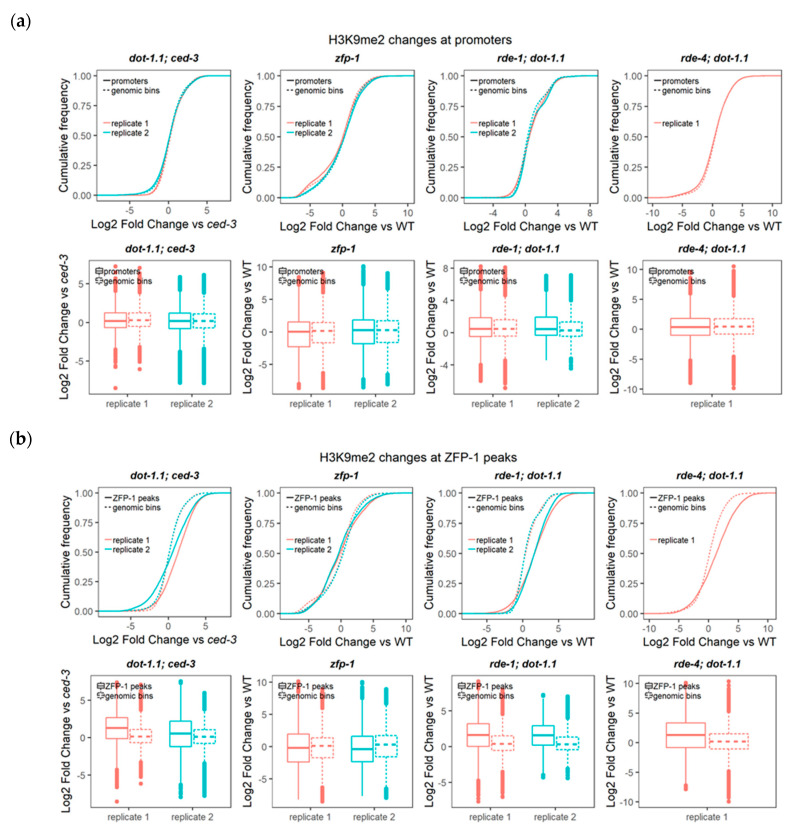
DOT-1.1 deletion does not affect the global H3K9me2 heterochromatin mark deposition at promoters, but rather at ZFP-1 peaks. Cumulative distribution of the fold change (log2) of H3K9me2 ChIP-seq reads per kilobase per million mapped (RPKM) values (input-normalized) between each mutant and the corresponding background strain at promoters (**a**) and ZFP-1 peaks (modENCODE data, GEO submission GSE50301 (**b**)). Two-sample Kolmogorov–Smirnov (for cumulative distribution plots) and Wilcoxon rank sum (for boxplots) tests were performed to compare the cumulative changes between each region set (either promoters or ZFP-1 peaks) and those in non-overlapping genomic bins spanning the genome. For ZFP-1 peak analyses, the statistical significance (*p*-values < 2.2 × 10^−16^) was found for all the *dot-1.1; ced-3*, *rde-1; dot-1.1* and *rde-4; dot-1.1* mutant replicates analyzed, as well as for the *zfp-1* mutant, with the exception of one of the replicates (Wilcoxon *p*-value = 0.2029).

**Figure 4 cells-09-01846-f004:**
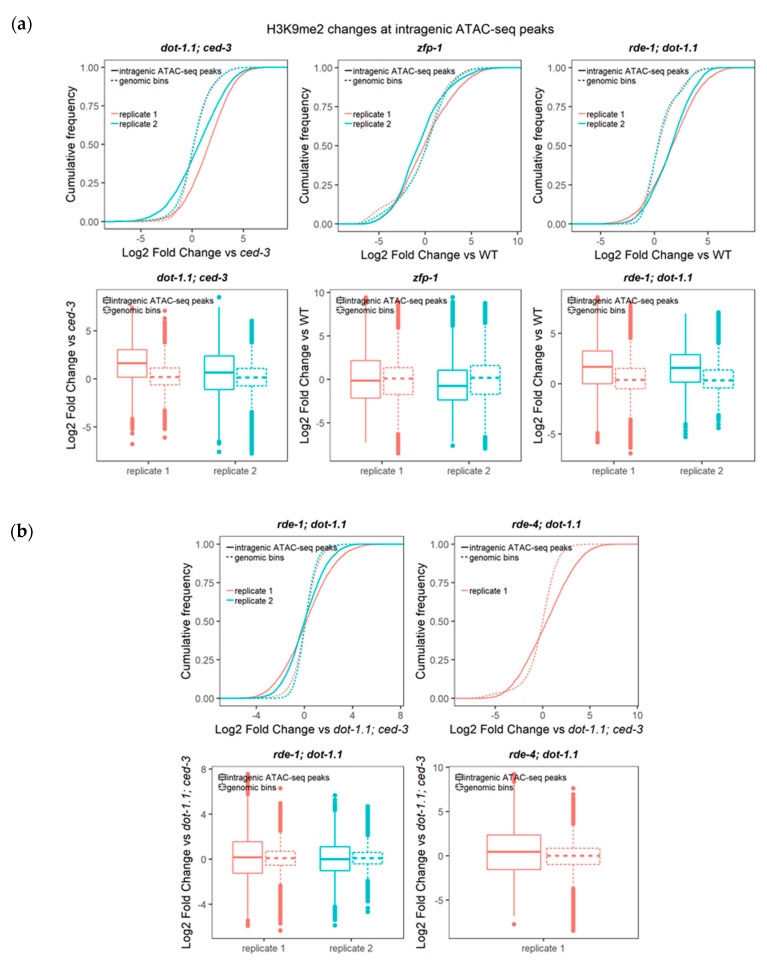
Deletion of DOT-1.1 leads to a global increase in H3K9me2 at ATAC-seq peaks. (**a**) Cumulative distribution of the fold change (log2) of H3K9me2 ChIP-seq RPKM values (input-normalized) at intragenic ATAC-seq peaks [47] between each mutant replicate and the average value in a background strain. Two-sample Kolmogorov–Smirnov (for cumulative distribution plots) and Wilcoxon rank sum (for boxplots) tests were performed to compare the cumulative changes between intragenic ATAC-seq peaks and those in non-overlapping genomic bins spanning the genome. Statistical significance (*p*-values ranging between < 2.2 × 10^−16^ and 3.67 × 10^−5^) was found for all the *dot-1.1; ced-3*, *zfp-1* and *rde-1; dot-1.1* mutant replicates analyzed. (**b**) Despite the significance (*p*-values ranging between < 2.2 × 10^−16^ and 0.0003988), the magnitude of changes in the comparisons between the *rde-1; dot-1.1* or *rde-4; dot-1.1* mutant strains with the *dot-1.1; ced-3* mutant is lower than that of the comparisons with the wild type (WT) strain (compare *rde-1; dot-1.1* in (**a**) and (**b**)).

**Figure 5 cells-09-01846-f005:**
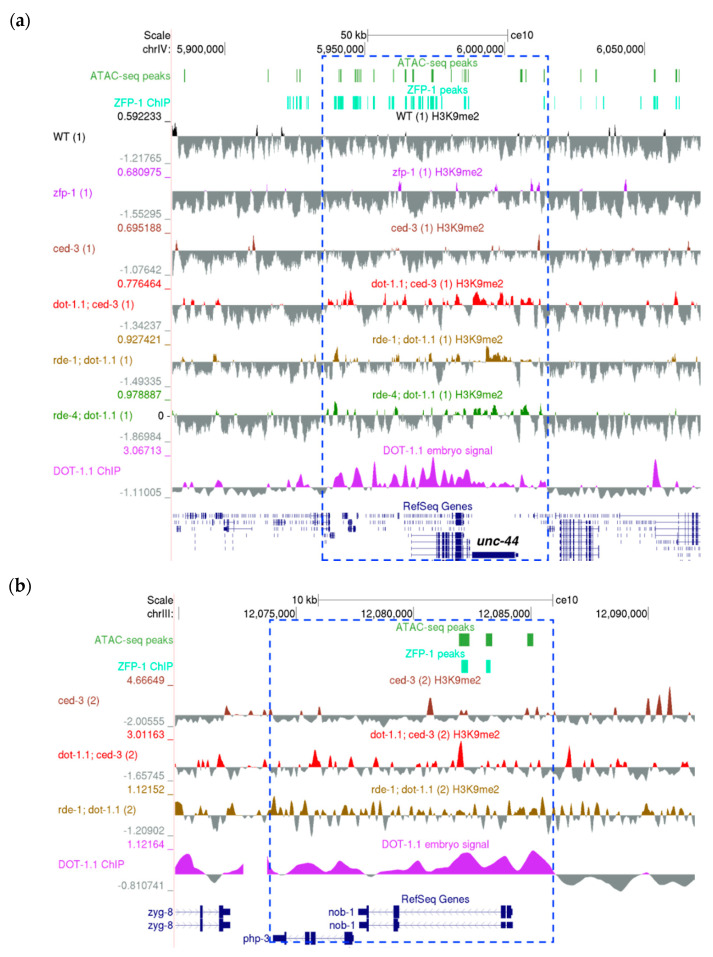
Examples of genomic loci containing developmental genes that gain H3K9me2 in the absence of DOT-1.1 (UCSC genome browser screenshots). (**a**) The >50kb genomic region bound by DOT-1.1 and containing the *unc-44* gene on ChrIV. (**b**) DOT-1.1 is present at the *C. elegans* homeobox (HOX) genes: the ~10kb *php-3/nob-1* locus on ChrIII. ATAC-seq peaks: [47]. ZFP-1 peaks: modENCODE data, GEO submission GSE50301. H3K9me2 coverage tracks: our data. DOT-1.1 ChIP-chip signal: modENCODE data, GEO submission GSE37488.

**Figure 6 cells-09-01846-f006:**
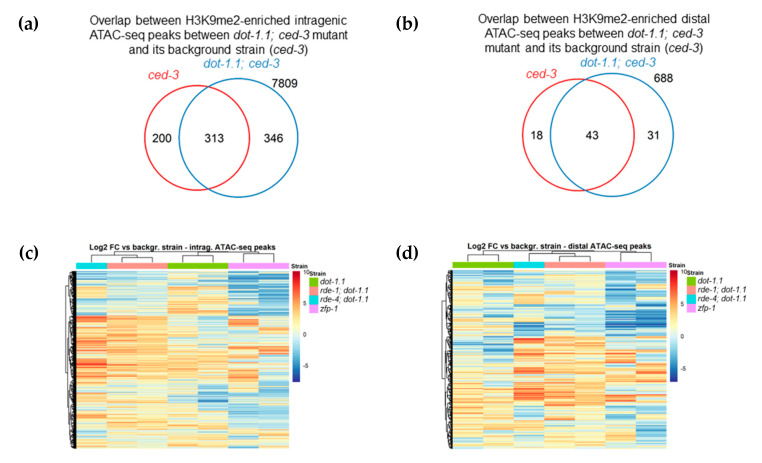
Venn diagram (**a**,**b**) and heatmap (**c**,**d**) representations of the H3K9me2 increases at ATAC-seq peaks in the *dot-1.1; ced-3* mutant compared with its background strain (*ced-3*). The Venn diagrams show that the *dot-1.1; ced-3* mutant has more intragenic (**a**) and distal (**b**) ATAC-seq peaks overlapping with H3K9me2 peaks (observed in both replicates, see Materials and Methods) than the background strain (*ced-3* mutant). Numbers inside the circles designate the ATAC-seq peaks overlapping with the H3K9me2 peaks in either strain alone (left-most and right-most numbers) or in both strains (middle). Numbers outside the circles represent the ATAC-seq peaks not overlapping with H3K9me2 peaks in any strain. The heatmaps show the fold changes (log2) of the H3K9me2 ChIP-seq RPKM value (input-normalized) between each mutant replicate and the average value in the corresponding background strain at intragenic (**c**) and (**d**) distal ATAC-seq peaks. Note that the values for the replicates within each mutant strain cluster together.

**Figure 7 cells-09-01846-f007:**
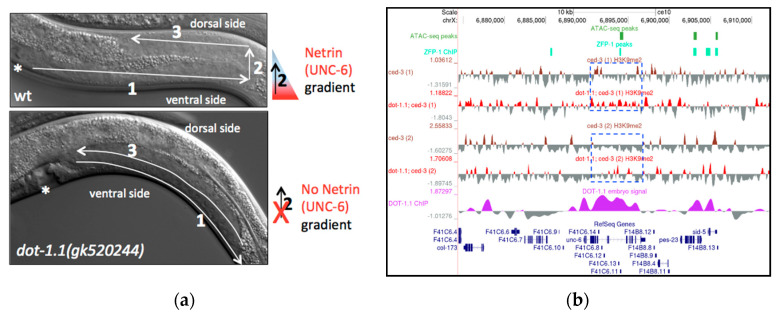
(**a**) Gonad migration defects in *dot-1.1* mutant worms resemble loss-of-netrin (UNC-6) signaling. Differential Interference Contrast (DIC) images: anterior to the left, posterior to the right. Posterior halves of L4 hermaphrodites are shown; vulva is denoted by an asterisk. Arrows indicate the three steps of gonad migration; in step 2, migration occurs dorsally, against the netrin (UNC-6) gradient. Top: in wild type worms, the gonad migrates away from the mid-body along the ventral side (step 1), then dorsally (step 2), and finally towards the mid-body along the dorsal side. Bottom: in *dot-1.1* mutants, the dorsal migration fails resulting in a “ventral” appearance of the gonad, typical of netrin signaling mutants. (**b**) A UCSC genome browser screenshot showing an increased H3K9me2 coverage at the *unc-6* locus, characterized by the presence of intronic enhancer signatures (ATAC-seq peak) in *dot-1.1(−)* replicate samples. ATAC-seq peaks: [47]. ZFP-1 peaks: modENCODE data, GEO submission GSE50301. H3K9me2 coverage tracks: our data. DOT-1.1 ChIP-chip signal: modENCODE data, GEO submission GSE37488.

**Figure 8 cells-09-01846-f008:**
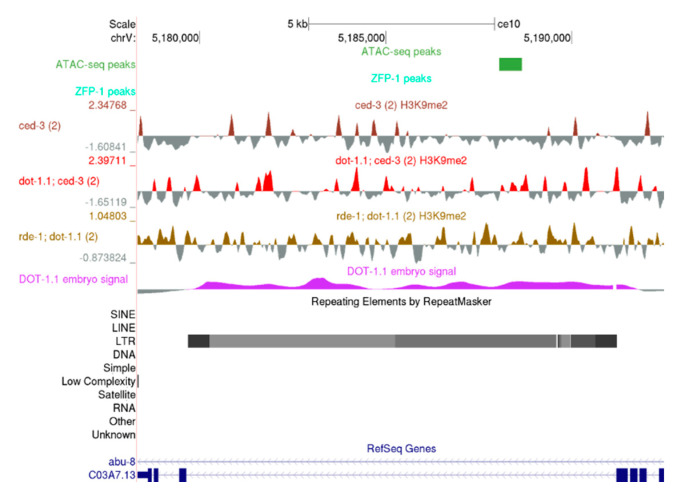
A UCSC genome browser screenshot showing increased H3K9me2 levels at Cer8 retrotransposon in *dot-1.1(−)* strains. ATAC-seq peaks: [47]. ZFP-1 peaks: modENCODE data, GEO submission GSE50301. H3K9me2 coverage tracks: our data. DOT-1.1 ChIP-chip signal: modENCODE data, GEO submission GSE37488.

**Figure 9 cells-09-01846-f009:**
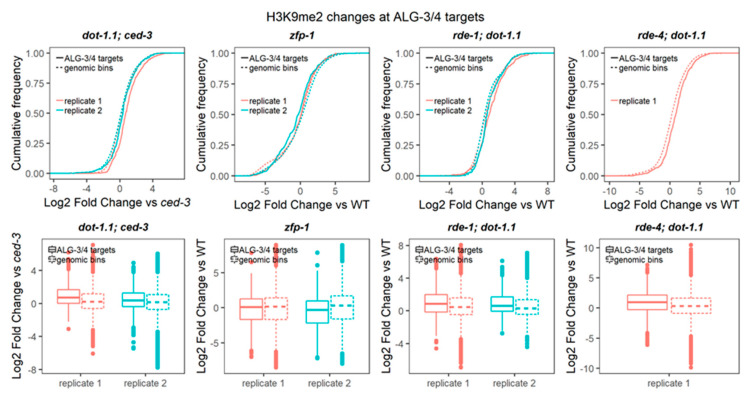
Deletion of DOT-1.1 leads to a global increase in H3K9me2 at targets of Argonautes, ALG-3 and ALG-4. Cumulative distribution of the fold change (log2) of H3K9me2 ChIP-seq RPKM value (input-subtracted) at ALG-3/4 targets [48] between each mutant replicate and the average value in the background strain. Two-sample Kolmogorov–Smirnov (for cumulative distribution plots) and Wilcoxon rank sum (for boxplots) tests were performed to compare the cumulative changes between ALG-3/4 targets and those in non-overlapping genomic bins spanning the genome. Statistical significance (*p*-values ranging between < 2.2 × 10^−16^ and 0.0008359) was found for all the *dot-1.1; ced-3*, *rde-1; dot-1.1* and *rde-4; dot-1.1* mutant replicates analyzed.

**Figure 10 cells-09-01846-f010:**
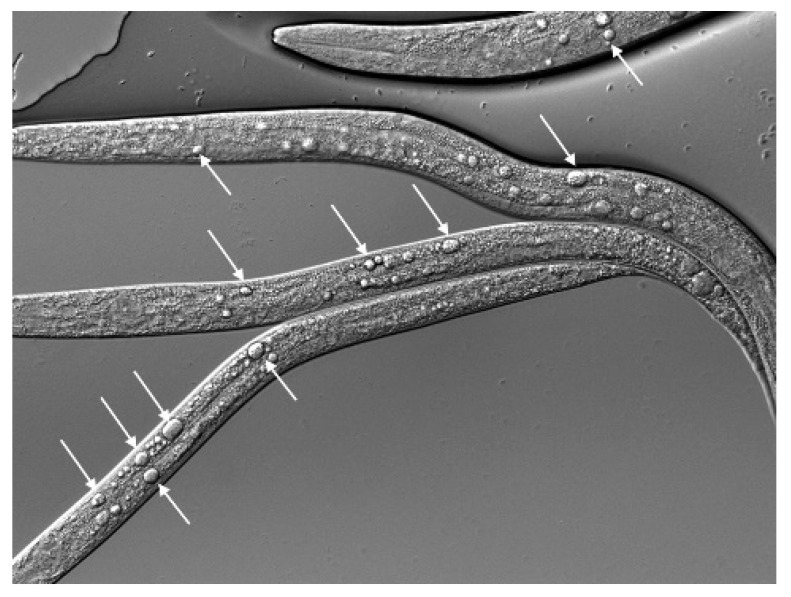
*dot-1.1(−)* animals do not survive in a wild type *ced-3(+)* background. A representative DIC image of *dot-1.1(−)* growth-arrested larvae. Arrows point to large vacuoles in the intestines.

**Table 1 cells-09-01846-t001:** Frequency of gonad migration defects in *dot-1.1* and *zfp-1* mutants.

Genotype	% with Phenotype	Total Gonads ^1^
Wild type (N2)	0	100
*dot-1.1(gk520244)*	44	166
*zfp-1(gk960739)*	5	232

^1^ The number of gonads is two per animal scored.

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
