# Peer review of "Caenorhabditis elegans Deficient in DOT-1.1 Exhibit Increases in H3K9me2 at Enhancer and Certain RNAi-Regulated Regions"

_cells, 2020, doi:10.3390/cells9081846_

Round 1

Reviewer 1 Report

In the manuscript “Caenorhabditis elegans deficient in H3K79 methylation exhibit increase in H3K9me2 at enhancers” Esse and Grishok analyze the genome wide changes in H3K9me2 upon loss, or reduction of H3K79me. The conclusions are based on H3K9me2 ChIPseq in C. elegans L3 larvae that are either wt, dot-1.1 mutant, a mutant that deletes the long isoform of zfp-1 and the rde-1/4;dot-1.1 double mutants which rescues lethality.

While the question is interesting and some of the conclusions do seem to be justified, we find that the types of analysis and plots used in the manuscript are not intuitive and do not always show all relevant information. Indeed, crucial bioinformatics comparisons and standardization of their normalization procedures are missing. Some conclusions need to be toned down (see below). A revised version should be re-reviewed prior to acceptance for publication.  

Specific points:

  1. Figure 1 correlates H3K9me2 and ZFP-1 ChIPseq enrichments either genome wide, or plots enrichment at enhancers. While the conclusions from Figure 1a are justified, the scatterplots show a potential problem in the normalization method used. The majority of the H3K9me2 signal appears to have a log2 FC – input of around -4 and ranges up to max. 2. ZFP-1 enrichment instead is concentrated at around -2 and ranges up to 2. While this does not affect the correlation, it results in a misleading representation of enrichments when only a fraction of the genome is plotted using the same scale for both IPs (like in Figure 1b). To correct this please adjust the parameter in the TMM normalization to achieve a similar “background” level between the IPs or plot a z-score.
  2. In section 3.1.2 an anti-correlation is proposed between H3K9me2 and ZFP-1 specifically at enhancers. The graph in Figure 1b however does not provide a measure of correlation.
  3. In Figure 1a the H3K9me2 enrichment spans from -4 to 2. In Figure 2a the H3K9me2 enrichment spans from -6 to 6, despite this being the same data (according to the figure legend). Please clarify which data is correct or how each was normalized. There needs to be better normalization of the data and/or an explanation of what was done.
  4. Figure 2b, cumulative plots: The cumulative plots used to compare changes in H3K9me2 levels between mutants are not intuitive and do not provide the full extent of information that a simple scatterplot, or the heat maps (as shown in Figure 1) can provide. In addition, the major conclusion is that H3K9me2 levels increase in dot-1.1 mutants specifically at enhancers. This comparison is however not represented in a graph. It would therefore be important to compare changes of H3K9me2 at e.g. repetitive elements, promoter and sites/ enhancers that do not contain a ZFP-1 signal. Basically the cumulative plots are very difficult to understand.
  5. In addition, it is important to ask whether the increase in H3K9me2 reflects an increase in already pre-existing H3K9me2, or the methylation of enhancers that previously lacked H3K9me2.
  6. In section 3.3 the authors conclude that the rescue of dot-1.1 lethality in the rde-1;dot-1.1 double mutant is not due to prevention of H3K9me2 at enhancers. I see two problems with this conclusion: 1) The authors do not provide a comparison between H3K9me2 in dot-1.1 and rde-1;dot-1.1. It is therefore not possible to judge whether a slight reduction (overall, or at specific enhancers) in the accumulation of H3K9me2 might be sufficient to change the enhancer activity. 2) If lethality is due to an aberrant enhancer activity in a specific tissue (e.g. the germline, or a neuron) then a H3K9me2 ChIPseq in a whole larvae would likely average out such a tissue specific event. Thus the conclusion is unwarranted.
  7. In the examples shown in Figure 5, 6 and 8 the H3K9me2 signal seems to be highly variable between replicas (at the chosen resolution), despite the relative decent correlation between replicas shown in Figure 1a. This is also the case in many of the cumulative plots. Please provide additional examples and comment on this variability – is it regional ?

Minor points:    

  1. On page 4 two sets of comparisons are mentioned a ced-3 and dot-1.1;ced-3 as well as the rde-1/4;dot-1.1. There is however no mention of a ced-3 or dot-1.1;ced-3 mutant in any of the results.  This is confusion and should be removed or additional data needs to be included.
  2. Please label axis of plots in Figure 1, 2a, 5, 6 and 8.
  3. Please expand your explanation of how enhancers were defined, as this is crucial in understanding the conclusions.

Reviewer 3 Report

The paper by Esse and Grishok investigates the interaction of H3K79 methylation and H3K9me in C. elegans.  Despite the early description of the role of Dot1 and H3K79me playing important roles in chromatin biology in yeast, little has been known about this methyl mark in higher metazoans.  This work nicely demonstrates the specific role of H3K9me in minimizing the spread of H3K9me2 on to enhancers.  The experiments in this paper are straight forward but add to the growing and important understanding of regulation of site specific regulation of H3K3me2 outside of the large heterochromatin domains.  It is very clearly written and presented and will be of broad interest to those who both work on chromatin and C. elegans development and gene expression.  My only major concern is the interpretation of the dot-1.1; rde-1 mutant ChIP data (see below) as there seems to be a difference between what is described in the text and what is shown in a number of figures.  Overall, this was a good paper that needs some small re-writes. 

Major Concerns:

My only major concern is that the interpretation of the difference in H3K9me2 enrichment over enhancers is the same/similar in level between dot-1.1 single mutants and rde-1; dot-1.1 or rde-4; dot-1.1 mutants.  In figures 3 and 4 there is a clear shift to the right in all of these combinations for H3K9me2 representing a increase in H3K9me2 at enhancers in dot-1.1 mutants whether or not there is loss of rde-1 or rde-3.  However; in the subsequent genome browser shots in figures 5, 6, and 8 there is little to no enhancement of H3K9me2 in the enhancer regions in the double mutants compared to the single mutants (especially rde-1; dot-1.1).  Because the reader is only presented with 5 examples of enhancers it isn’t clear how representative these genome browser shots are, but it seems like the interpretation that there is no RNA role in mediating the deposition of H3K9me2 may be overstated or wrong at least for the presented loci.  This difference between these specific genes on the genome browser and the genome-wide data beg the question of if the enhancers that have enhanced H3K9me2 in dot-1.1 and rde-1;dot-1.1 are the same.  There is no indication that the authors investigated this but it would strengthen their interpretation of the data.

In addition, there needs to be a discussion of why the H3K9me2 that is enriched at enhancers in dot-1.1 mutants shows very different patterns between replicates.  What do the authors think underlie these differences?

Minor concerns:

  1. The figure legends are very short and it would help readers if they were expanded somewhat to more clearly describe what is shown in the figures.

  1. Why in figure 2A is there not a corresponding figure with the H3K9me2 coverage in the dot-1.1 Maybe they don’t need to show all six chromosomes but could just give examples in WT and mutants.  The graphs shown in Figure 2B do not rule out that although there is similar levels of H3K9me2 in the mutants and WT that these are have gross differences in localization which can only be seen with the first visualization.

  1. There needs to be more explained in the materials and methods for a number of points:
  • The method used in Figure 1 should be in the materials and methods- not just the figure legend.
  • The method used to make the plots in Figure 2B, Figure 3 and Figure 4 should be in the materials and methods in addition to specifics about the statistics shown in the figure legend for Fig 3 and Fig 4.
  • A description of the method for scoring the migration (how were the images actually scored for if the migration was done correctly or not) should be included in the materials and methods.

Round 2

Reviewer 4 Report

The authors have addressed all of my concerns.  thank you.

Author Response

We have made changes to correct minor spelling mistakes and proofread the manuscript.